# Neural circuitry of social learning in Drosophila requires multiple inputs to facilitate inter-species communication

Balint Z. Kacsoh [1], Julianna Bozler[1], Sassan Hodge[1] & Giovanni Bosco[1]

Drosophila species communicate the threat of parasitoid wasps to naïve individuals. Communication of the threat between closely related species is efficient, while more distantly related species exhibit a dampened, partial communication. Partial communication between *D. melanogaster* and *D. ananassae* about wasp presence is enhanced following a period of cohabitation, suggesting that species-specific natural variations in communication 'dialects' can be learned through socialization. In this study, we identify six regions of the Drosophila brain essential for dialect training. We pinpoint subgroups of neurons in these regions, including motion detecting neurons in the optic lobe, layer 5 of the fan-shaped body, the D glomerulus in the antennal lobe, and the odorant receptor Or69a, where activation of each component is necessary for dialect learning. These results reveal functional neural circuits that underlie complex Drosophila social behaviors, and these circuits are required for integration several cue inputs involving multiple regions of the Drosophila brain.

---

[1] Department of Molecular and Systems Biology, Geisel School of Medicine at Dartmouth, Hanover, NH 03755, USA. Correspondence and requests for materials should be addressed to B.Z.K. (email: balintzkacsoh@gmail.com) or to G.B. (email: giovanni.bosco@dartmouth.edu)

The ability to decipher and react to environmental information is a phenomenon intrinsic to all advanced life forms. Organisms can benefit by acquiring relevant information about the environment either through direct experience or by way of second-hand information from others; this type of information transfer has been observed in a wide range of taxa and can occur within and between species. Such information exchange falls under the concept of 'perceptual learning'—the process by which the ability of an organism to respond to environmental cues is improved through experience. Although the line that distinguishes transfer of information from one individual to another and what constitutes communication is not clear, the study of animals that change their ability to receive information based on both social interactions and past experiences with specific groups may help further our understanding of the neuronal basis of communication. Social communication, specifically, is most extensively documented in complex organisms, such as mammals and birds; however, insects can also display a broad range of sociality, and in some cases, communication is an inherent property of their social structure. When one observes insect groups in nature, their behavior readily reveals distinct planes of cooperation, communication, and social structure[1].

Eusocial insects have been the traditional model for this type of study; however, non-caste based insects offer a unique opportunity to probe questions regarding social structure. Recent studies have indicated that once canonically 'solitary' fruit fly can be utilized to study rudimentary social interactions, including social learning[2]. Oviposition (egg laying) site and food selection are also driven by social exchange mediated by olfactory cues and neural processing[3], in addition to the condition of the environment itself[4]. Recent work has also suggested the importance of the communal aspect of the Drosophila lifecycle involving tumor genesis, where Drosophila are able to discriminate between individuals at different stages of tumor progression and prefer social environments of individuals without tumors[5]. It has also been suggested that social learning in Drosophila can lead to persistent mate choice reminiscent of cultural transmission, and that such choices can in turn be inherited through social learning from one generation to the next[6,7]. Thus, there is mounting experimental evidence that Drosophila could be useful as models for understanding genetic and neurophysiological mechanisms of social learning as well as evolution of social structures, providing us with important clues as to how more complex behavior may have evolved.

Adult Drosophila across the genus have been in an arms race against predators and have subsequently evolved complex immune and behavioral changes to protect their offspring from predators such as endoparasitoid wasps, which can prey on immature stages of larvae of certain Drosophilid species[8,9]. The behavioral changes of adult Drosophila in response to wasps include altered food preference and reduced oviposition (egg laying)[10–16]. The wasp predator threat also triggers social transference of information between experienced and naive individuals. Wasp-exposed teacher flies communicate the wasp threat to naive flies through visual cues, which then depress their own oviposition rate[13]. Communication of the predator threat can occur within a species, but it also occurs between related species. Closely related species demonstrate an efficient level of communication. A dampened, or partial communicative ability is observed when more distantly related species are paired, while very distantly related species lack the ability to communicate all together[15]. Remarkably, the observed partial communication between some species is alleviated following a cohousing period, termed "dialect training," where an exchange of visual and olfactory signals is enabled. This observation is highly suggestive

of natural variations in modes of communication, similar to the evolution of linguistic variations between species, termed "dialects"[15–17]. Thus, a Drosophilid can expand is communication repertoire by learning another species' dialect, through dialect training. Moreover, these observations also indicate that barriers that normally may hinder efficient communication may be overcome through socialization, suggesting that plasticity within specific regions of the brain and previous social experience are both important for efficient adult communication.

In the present study, we focus on neural circuitry of dialect training and dialect acquisition in the Drosophila system: Inter-species dialect training, unlike intraspecies social learning, is multi-modal, requiring, at minimum, olfactory, visual, sex-specific, and temporal cues for successful dialect acquisition. This complexity provides the unique opportunity to probe a multi-level neural circuit directing in social interaction[13,15]. Using this fly–fly social learning paradigm, we asked (1) what neuronal groups are required for inter-species dialect training and acquisition and (2) what is the subset of neurons/receptors in a given neuronal structure that is necessary for dialect learning.

## Results

**Dialect training is governed by multiple brain regions.** In order to quantify communication ability, we utilized the fly duplex; an apparatus with two transparent acrylic compartments allowing flies to see other flies or wasps in the adjacent compartment, without direct contact (Fig. 1a). For detailed experimental design see the Methods section. In brief, ten female and two male D. ananassae are placed into one duplex compartment, with an adjacent compartment containing 20 female wasps (or no wasp for control). Following a 24-h exposure, wasps are removed and acute response is measured by counting the number of eggs laid in the first 24-h period in a blinded manner. Here, an acute response is defined as the number of eggs laid in the presence of wasp or 'during wasp exposure'. Flies are shifted to a new duplex after the 24-h exposure, with ten female and two male D. melanogaster naive student flies in the adjacent compartment (Fig. 1a, see Methods section). After this second 24-h period where teacher and student flies are allowed to interact, all flies are removed and the response of both teacher and student is measured by counting the number of eggs laid in a blinded manner (see Methods). The 24–48-h period measures memory of teachers having seen the wasps and students having learned from the experienced teachers. As previously shown, using wild-type D. ananassae, we find both an acute response and a memory response to the wasp in teacher flies, and a partial social learning response in untrained D. melanogaster student flies (Supplementary Fig. 1, Supplementary Data 1 for all raw egg counts and p values)[10,13,15,18]. We replicated a previous finding demonstrating the ability of D. melanogaster to enhance its communication ability with D. ananassae following a weeklong cohousing termed "dialect training" (Supplementary Fig. 1)[15]. This dialect training period enhances communication as shown by students paired with wasp-exposed teachers (Fig. 1c, d). Using this paradigm, we wished to elucidate the neural circuitry that governs this complex behavior by deactivating defined subsets of neurons in an inducible fashion. Understanding dialect training, memory storage, and subsequent retrieval requires a fundamental knowledge of the underlying neuronal circuits, currently unknown.

In order to identify brain regions that govern dialect learning, we utilized the FlyLight database, from which we selected GAL4 drivers that marked singular structures strongly and ubiquitously[19]. The FlyLight GAL4 lines were generated by insertion of defined fragments of genomic DNA, serving as transcriptional enhancers. We selected nine brain structures to analyze: the optic

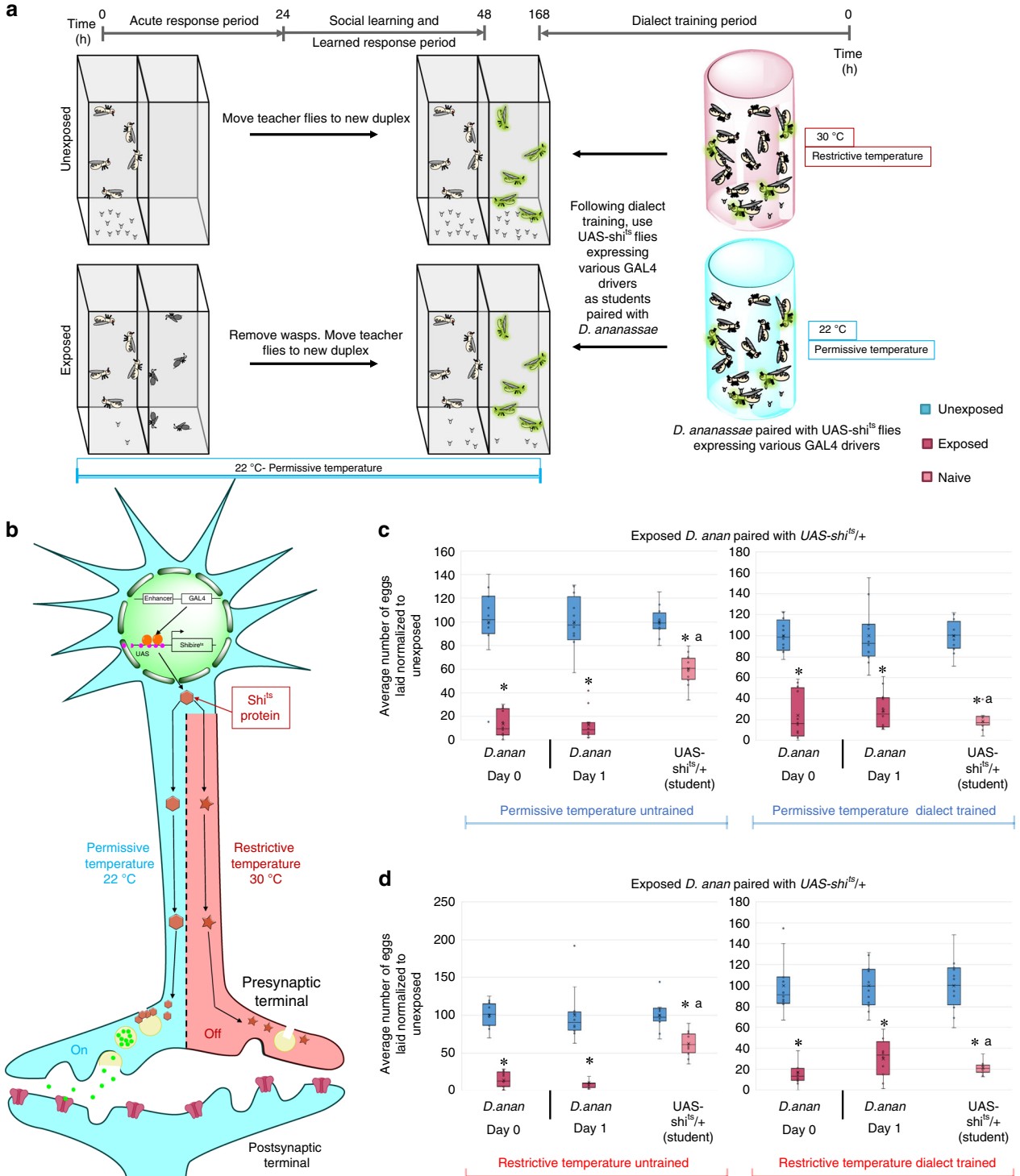

**Fig. 1** Dialect training is governed by multiple brain regions revealed by UAS-shi[ts]. **a** Standard experimental design using UAS-shibire (UAS-shi[ts]) in conjunction with various GAL4 drivers. Dialect learning/training is performed at either the permissive (22 °C) or restrictive (30 °C) temperature, while the wasp exposure and social learning period is performed exclusively at the permissive temperature. **b** Schematic of the UAS-shi[ts] expression in neurons, where the restrictive temperature turns off the neurons of interest and the permissive temperature does not affect these neurons. **c** Percentage of eggs laid by exposed flies normalized to eggs laid by unexposed flies is shown. UAS-shi[ts] outcrossed to *Canton S* trained by *D. ananassae* at either the permissive temperature **c** or the restrictive temperature **d** show wild-type dialect acquisition in both the naive and trained states. Blue lines indicate permissive temperature, while red lines indicate restrictive temperature. Trained and untrained states are labeled. Plots are standard Tukey Box Plots, where the bounds of the shaded box represent the first and third quartile; whiskers indicate the minimum and maximum data point within 1.5 times the interquartile range. Mean is denoted with an 'X'. Data used to generate the plots have been superimposed on the graph as open circles ($n = 12$ biologically independent experiments) (*$p < 0.05$, a → $p < 0.05$ comparing trained and untrained students within a given temperature when both restrictive and permissive temperatures are tested)

lobe, mushroom body, antennal lobe, lateral horn, fan-shaped body, ellipsoid body, prow, superior clamp, and the bulb[20–22]. If available, we obtained two unique driver lines matching our selection criteria. Using mCD8-GFP as a reporter, immunofluorescence reveals GFP expression constrained to these selected regions of interest, validating these lines for testing (Supplementary Figs 2–14a, b). It remains important to note that any negative findings may be the result of the sensitivity and cell numbers marked of these constructs and that future testing of additional lines may prove beneficial.

These GAL4-driver lines were employed to drive the expression of the temperature sensitive dominant negative dynamin *shibire*, UAS-shibire[ts] (UAS-shi[ts]). This method allowed us to temporarily deactivate particular neurons of interest during dialect training by shifting the animals to a higher temperature compartment; importantly this neuron-deactivation is reversible at the 'permissive' temperature[23]. Dialect training was performed at either the restrictive (30 °C) or permissive (22 °C) temperature, after which dialect-trained or untrained *D. melanogaster* were paired with either unexposed or wasp-exposed *D. ananassae* teachers at the permissive temperature (Fig. 1a). This experimental setup ensures that the UAS-shi[ts] construct is functional only during the dialect training period (Fig. 1b). Wild-type flies show normal dialect-trained and untrained states when incubated at either 22 or 30 °C for 1 week, indicating temperature shifts alone during the cohabitation period do not adversely affect dialect training (Supplementary Fig. 1). In addition, outcrossed UAS-shi[ts] (UAS-shi[ts]/+) show wild-type untrained and dialect-trained states at both the restrictive and permissive temperatures, demonstrating that perturbation of dialect learning is likely not due to genetic background of the UAS-shi[ts] construct (Fig. 1c, d).

Previous results have indicated that visual system and mushroom body (MB) are required for dialect learning[15,24]. Recent work has highlighted direct neural pathways that convey visual information from the optic lobe to the MB, demonstrating a possible connectivity of the two circuits[25]. As a proof of concept, we tested two optic lobe and MB drivers in conjunction with UAS-shi[ts]. When dialect training is performed at the restrictive temperature, both optic lobe and MB GAL4 lines driving expression of UAS-shi[ts] result in perturbed dialect learning (Supplementary Figs 2–5). This was true for two unique GAL4-driver lines for each MB and optic lobe regions, while at the permissive temperature flies exhibited wild-type dialect learning (Supplementary Figs 2–5). These results validate our experimental approach, and further suggest that dialect training is partially by visual inputs and MB-dependent learning and memory circuitry.

Additional GAL4-mediated expression of UAS-shi[ts] identifies four additional brain regions governing dialect learning, where we observe perturbed dialect training at the restrictive temperature, but not at the permissive temperature: the antennal lobe (Supplementary Fig. 6 and 7), lateral horn (Supplementary Fig. 8), fan-shaped body (Supplementary Fig. 9), and ellipsoid body (Supplementary Figs 10 and 11). We find three regions, the bulb (Supplementary Fig. 12), prow (Supplementary Fig. 13), and superior clamp (Supplementary Fig. 14), whose activation is dispensable for dialect training when tested with the genetic reagents available. Future studies with different signal patterns in these structures, limited to fewer/other neurons may provide unique results. In addition, the UAS-shi[ts] construct used in this study may have cell-to-cell variability that could generate negative results. Collectively, we identify six regions of the brain that require activation during dialect training. The finding that three structures can be inactivated and still permit dialect learning suggests that inactivation of any brain structure is not sufficient to perturb dialect training. This finding further suggests that the regions identified are important and are appropriate candidates for further circuit dissection.

**Dialect training is partially mediated by Or69a.** Following the observation that the antennal lobe is required for dialect training (Supplementary Figs 6 and 7), we wished to identify a more precise series of neurons and possible pheromone receptor candidates involved in dialect training.

The Drosophila olfactory system is comprised of four sensillum classes—antennal trichoids, antennal basiconics, antennal coeloconics, and palp basiconics. These detectors are present on the third antennal segment and the maxillary palp (Fig. 2a). Housed within these sensillar classes are olfactory sensory neurons (OSNs), of which there are ~1300, that project from sensilla to innervate the glomeruli of the antennal lobe (Fig. 2a)[26,27]. Olfactory glomeruli are morphologically conserved, spherical compartments of the olfactory system located in the antennal lobe. These 52 glomeruli are distinguishable by their chemosensory repertoire, position, and volume. Glomeruli receive information regarding olfactory detecting from responses originating from OSNs in the sensilla. Individual OSNs express only one receptor gene in conjunction with the *Orco* co-receptor, providing specificity. Importantly, *Orco* function is necessary for dialect training[15]. Detected information is then subsequently relayed to other regions of the brain (i.e., to the mushroom body signaling to the lateral horn by projection neurons).

We leveraged the current knowledge of the Drosophila olfactory system to identify potential olfactory receptor candidates that might be involved in dialect learning. In particular, we identify Or47b, Or65a, and Or69a as potential candidates, which have been shown to play a role in other social interaction based assays[28–30]. The glomeruli enervated by each of the Ors respective OSNs are also marked in both antennal lobe drivers we utilized previously, making them appropriate candidates for testing (Supplementary Figs 6 and 7).

The sensillum that houses the Or47b receptor enervates the VA1lm glomerulus of the Drosophila antennal lobe (Supplementary Fig. 15a, c)[31]. Previous studies have shown that Or47b receptor and receptor neurons are necessary for social–sexual interactions, such as locomotor activity and nocturnal sex drive[29]. However, we find that flies mutant in Or47b (Or47b[−/−]) and RNAi (Or47b[RNAi]) expressing *D. melanogaster* targeting Or47b using an Or47b-GAL4 results in wild-type dialect training, suggesting that dialect learning is not directly mediated by Or47b (Supplementary Fig. 15b, d).

The sensillum that houses the Or65a receptor enervates the DL3 glomerulus of the Drosophila antennal lobe (Fig. 2a, Supplementary Fig. 15e, g)[31]. Previous studies have shown that the Or65a receptor and receptor neuron is involved in mate recognition and other complex social behaviors in male and female flies[28,32–34]; therefore making it an interesting candidate for dialect training. Yet, we find that flies mutant in Or65a (Or65a[−/−]) and RNAi (Or65a[RNAi]) expressing *D. melanogaster* targeting Or65a using an Or65a-GAL4 results in wild-type dialect training, suggesting that dialect learning is not directly mediated by Or65a (Supplementary Fig. 15f, h).

Or69a is expressed in the third antennal segment and is involved in long-range, species-specific pheromone detection as well as food detection[30]. The sensillum that houses the Or69a receptor enervates the D glomerulus of the Drosophila antennal lobe via the ab9A OSN (Figure 2a–d, Supplementary Fig. 16a)[22,31]. The compound detected by this receptor is the cuticular hydrocarbon *Z4-11A1* that acts as a species identifier in both sexes. Or69a also binds kairomonal terpenoids, such as linalool or terpineol, which are found in both fruit and yeast

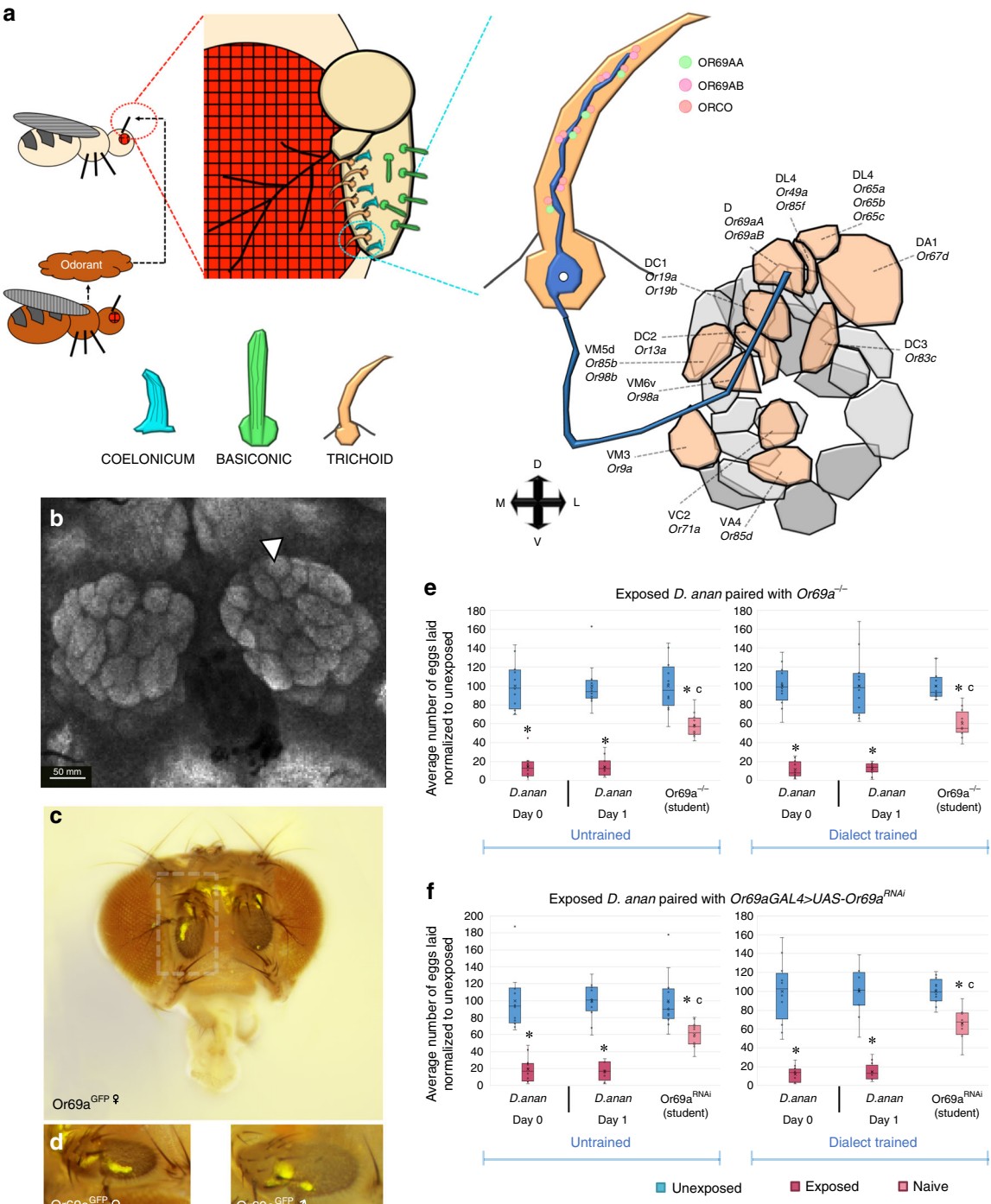

**Fig. 2** Dialect training mediated by the odorant receptor Or69a. **a** Schematic of *Drosophila* antenna which has three various olfactory detecting centers: coelonicum, basiconic, and trichoid. Or69a is highlighted in the closeup and its enervation of the D glomerulus in the antenna lobe is highlighted. Glomeruli in the same focal plane are colored orange, while glomeruli in a different focal plane are colored shades of gray. **b** The *Drosophila* antennal lobe with the D glomerulus indicated with an arrowhead. Nc82 staining shown in gray. **c**, **d** Expression of Or69a using the Or69a^GFP construct highlighting expression in the antenna. Represented in (**c**) is a female fly head, while (**d**) represents magnified antennae (shown in box) of both female and male flies expressing Or69a^GFP. **e** Percentage of eggs laid by exposed flies normalized to eggs laid by unexposed flies is shown. Or69a$^{-/-}$ show wild-type untrained behavior, but are unable to learn the dialect from *D. ananassae* following training. **f** Or69a^GAL4 driving Or69a^RNAi shows wild-type untrained behavior, but is unable to learn the dialect from *D. ananassae* following training. Blue lines indicate permissive temperature was used in all experiments. Trained and untrained states are labeled. Plots are standard Tukey Box Plots, where the bounds of the shaded box represent the first and third quartile; whiskers indicate the minimum and maximum data point within 1.5 times the interquartile range. Mean is denoted with an 'X.' Data used to generate the plots have been superimposed on the graph as open circles ($n = 12$ biologically independent experiments) (*$p < 0.05$, c → $p = $ ns comparing trained and untrained students in experiments with no restrictive temperature)

headspace, suggesting food and oviposition site detection[30]. We find that flies mutant in Or69a (Or69a$^{-/-}$) and RNAi (Or69a$^{RNAi}$) expressing *D. melanogaster* targeting Or69a using an Or69a-GAL4 results in perturbed dialect training (Fig. 2e, f). Outcrossed Or69a mutants (Or69a$^{+/-}$), in addition to UAS (UAS-Or69a$^{RNAi}$/+) and GAL4 (Or69a-GAL4/+) lines show wild-type dialect learning (Supplementary Fig. 16b–d), suggesting that our results are not due to genetic background effects. Our findings indicate that Or69a is essential for dialect acquisition.

This in-depth analysis into the role of antennal lobe neural circuitry and its associated olfactory receptors in dialect learning reveals the role of Or69a as an olfactory receptor involved in dialect training during communal living. Or69a has been previously demonstrated to have a dual affinity for both sex and food odorants, in a species-specific manner. Our data provide further evidence of the complex integration of pheromonal cues in Drosophila, involving integration of socially and environmentally derived cues. We also rule out Or47b and Or65a receptors, both previously indicated to be involved in other social interactions. Finally, we identify the OSN that enervates the D glomerulus in the antennal lobe as part of the neural circuit governing dialect learning.

**Dialect training is mediated by motion-detecting circuitry.** Given the observation that the optic lobe is necessary for dialect training, we wished to further dissect this brain region. In Drosophila, motion detection requires synaptic outputs of the R1–R6 photoreceptors. These photoreceptors project their axons into the first optic neuropil, known as the lamina. This projection forms a retinotopic map of visual space[35,36]. The visual map is comprised of 800 columnar elements, within which R1–R6 make synaptic connections with three projection neurons: L1, L2, and L3, as well as the local interneuron amc (Fig. 3a)[37,38]. Each of these optic lobe neurons has been shown to respond to different types of inputs. The L1 neuron has been shown to provide input to a pathway that detects moving light edges, while the L2 neurons provide input into a pathway that detects moving dark edges[39–41]. The L3 neuron is thought to inform landmark orientation and spectral preference, in addition to moving dark and light edges[42]. L1–L3 neurons represent all of the direct second-order relays from R1 to R6 into the medulla. Interestingly, L2 makes synaptic contacts with a third order monopolar cell, L4, which has been proposed to be involved in motion sensing, though not experimentally shown (Fig. 3a)[43–45].

Previous data suggest that a moving visual cue, provided by wing movement, is detected by *Drosophila* during the dialect-training period[15]. Given this observation, we wondered whether motion-detecting neurons in the optic lobe could be responsible for acquisition of visual information dialect learning. To explore this, we again utilized the GAL4 UAS-shibire$^{ts}$ (UAS-shi$^{ts}$) system, as described previously. We tested the L1-, L2-, L3-, and L4-GAL4 lines with strong, but constrained, expression patterns (Fig. 3b, d, f, h)[45,46]. Silencing of the L1 or L3 neurons resulted in wild-type dialect training (Fig. 3c, g), indicating that activation of the L1 or L3 neurons is dispensable for this behavior, when using this genetic reagent. However, silencing of the L2 motion-detecting neuron resulted in the inability of *D. melanogaster* to learn the dialect, suggesting that L2 neurons are necessary for this behavior (Fig. 3e). L4 receives most of its synaptic inputs from L2, but is also interconnected with neighboring dorso- and ventroposterior cartridges[38]. When we silenced the L4 neurons, we find that the ability of *D. melanogaster* to learn the dialect is perturbed, suggesting that they are necessary for this behavior (Fig. 3i). To further validate this result, we tested a *splitL4*$^{GAL4}$ line (L4$^{0980}$-VP16AD, L4$^{0987}$-GAL4DBD), which has a more

constrained expression pattern than the L4$^{0980-GAL4}$ alone (Supplementary Fig. 18); in conjunction with UAS-shi$^{ts}$, we find that the ability of *D. melanogaster* to learn the dialect is again hindered (Supplementary Fig. 18). Our finding uncovers a novel role for the L4 neurons, which have been suggested to be involved in motion detection in spatial summation by anatomical experimentation, but have never been implicated in a specific behavioral paradigm[45]. All non-dialect-trained animals at the control and restrictive temperatures behave as wild-type (Supplementary Fig. 17 and 18).

Collectively, our genetic dissection of the optic lobe neurocircuitry and its involvement in dialect training reveals that a visual stimulus is detected during dialect learning primarily by the dark edge detecting L2 circuit that enervates the L4 neuron, where activation these neurons are necessary for dialect learning (Fig. 3).

**Dialect training is mediated by region 5 of the fan-shaped body.** Given the observation that the fan-shaped body is necessary for dialect training (Supplementary Fig. 9), we wished to identify a more precise neuron subset involved in this process. The fan-shaped body (FB) is a member of the central complex[47], the FB is reported to be important for multiple functions, including control of locomotion[48], visual feature recognition[49] and visual information processing[50], courtship maintenance[51], and sleep regulation[51–54]. The FB has been also implicated in the regulation of electric- and heat-shock-induced innate avoidance and conditioned avoidance[55]. The FB is composed of at least nine layers making the neuropil an interesting target for further dissection[55,56].

We first focused on the entire FB, followed by subsets of FB neurons, by selecting GAL4 lines from the FlyLight database[19]. The lines were chosen based on covering either the entirety (R75G12) or a subset (R38E07, R89E07, R49H02) of the neurons in the FB, while labeling few to no other regions of the brain. Using mCD8-GFP as a reporter, immunofluorescence reveals that all FB regions are labeled by R75G12 (Fig. 4b); regions 5, 8, and 9 labeled by R38E07 (Fig. 4d); regions 2, 8, and 9 by R89E07 (Fig. 4f); and regions 1, 4, and 6 by R49H02 (Fig. 4h). We individually blocked the output of these neurons by expression of UAS-shibire$^{ts}$ (UAS-shi$^{ts}$) at the restrictive temperature. We find that inactivation of the entire FB, or regions 5, 8, and 9 with the R38E07 driver line, result in perturbed dialect training at the restrictive temperature, yet these flies display wild-type dialect training at the permissive temperature (Fig. 4c, e). The untrained flies demonstrate wild-type communication at both restrictive and permissive temperatures (Supplementary Fig. 19a–d). Inactivation of regions 2, 8, and 9 with the R89E07 driver line and inactivation of regions 1, 4, and 6 with the R49H02 driver line result in wild-type dialect training in both the restrictive and permissive temperatures, and wild-type communication in untrained states, suggesting that activation of these regions are not necessary for dialect learning (Fig. 4g, i, Supplementary Fig. 19e–h). Therefore, we conclude that region 5 of the FB is necessary for dialect learning, whereas regions 1, 2, 4, 6, 8, and 9 are dispensable, when using these particular genetic reagents. It remains possible that these other regions of the FB may need to be inactivated during dialect training and thus, their role would not be detected using this experimental approach.

## Discussion
In this study, we have elucidated the fundamental neural circuitry that governs dialect learning in Drosophila. When a Drosophild is placed next to a wasp-exposed Drosophilid of the same species (intraspecies), effective communication occurs, where wasp-exposed teachers inform naive students of a wasp threat,

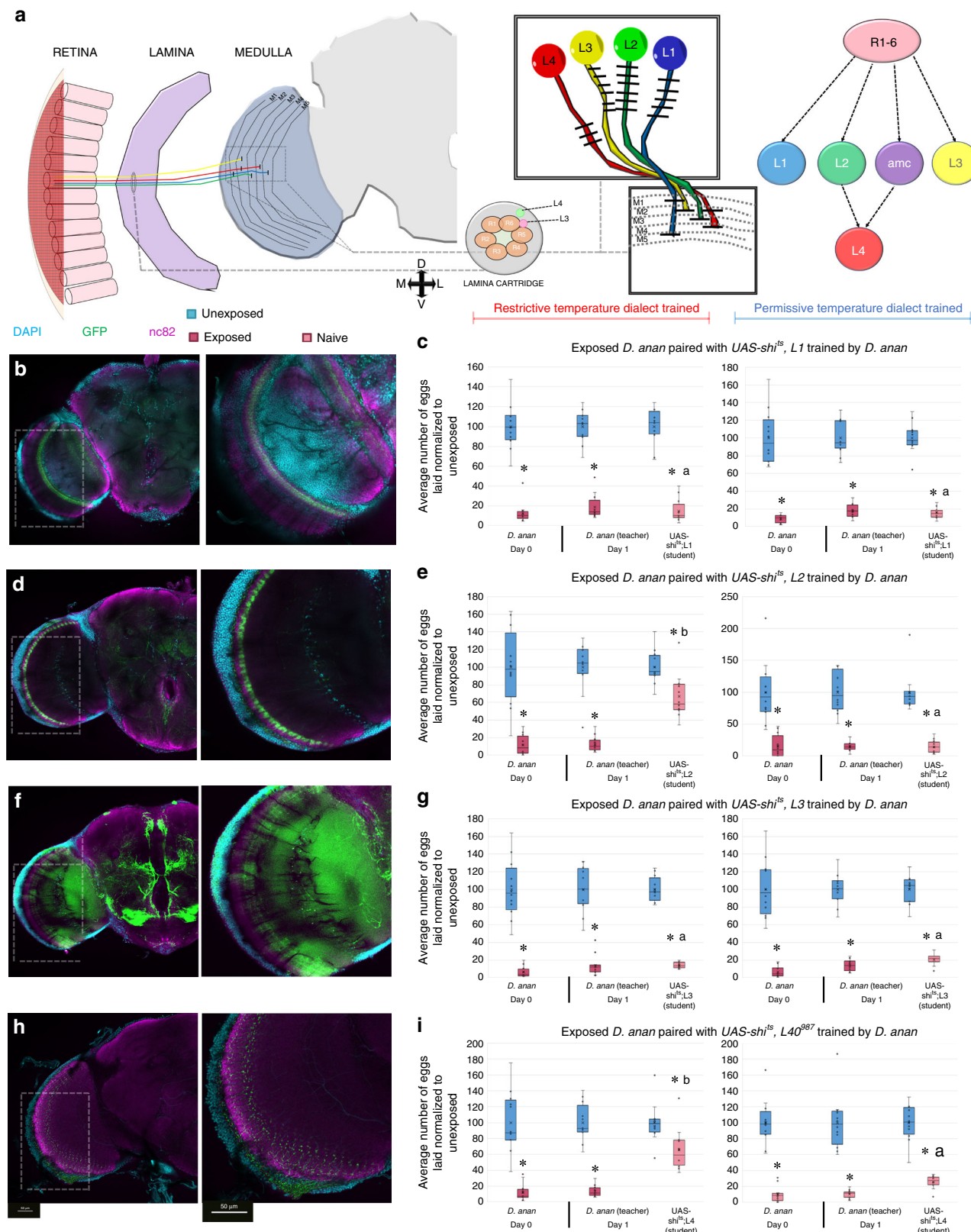

resulting in both teacher and student flies laying fewer eggs[15]. When naive *D. melanogaster* students are placed next to wasp-exposed *D. ananassae*, *D. melanogaster* students exhibit only partial communication ability. *D. melanogaster* can learn the dialect of *D. ananassae* after a period of cohabitation (dialect learning/training), yielding inter-species communication

enhanced to levels normally observed among conspecifics. Using the FlyLight library, in conjunction with UAS-shi[ts] to inducibly deactivate defined subsets of neurons during dialect training, we identify six distinct regions of the fly brain that are involved in dialect training: the optic lobe, mushroom body, antennal lobe, ellipsoid body, fan-shaped body, and the lateral horn[19]. We also

**Fig. 3** Motion-detecting neurons in the optic lobe are required for dialect training. **a** Schematic of *Drosophila* visual system. The optic lobe is shown highlighting the L1, L2, L3, and L4 neurons. Dialect learning is performed at either the permissive (22 °C) or restrictive (30 °C) temperature, while the wasp exposure and social learning period is performed exclusively at the permissive temperature. **b** Confocal image of adult brain where L1GAL4 is driving UAS-CD8-GFP, stained with nc82 (magenta) and DAPI (teal). **c** UAS-shits crossed to L1GAL4 trained by *D. ananassae* at the restrictive (left) and permissive (right) temperature. **d** Confocal image of adult brain where L2GAL4 is driving UAS-CD8-GFP, stained with nc82 (magenta) and DAPI (teal). **e** UAS-shits crossed to L2GAL4 trained by *D. ananassae* at the restrictive (left) and permissive (right) temperature. **f** Confocal image of adult brain where L3GAL4 is driving UAS-CD8-GFP, stained with nc82 (magenta) and DAPI (teal). **g** UAS-shits crossed to L3GAL4 trained by *D. ananassae* at the restrictive (left) and permissive (right) temperature. **h** Confocal image of adult brain where L40987-GAL4 is driving UAS-CD8-GFP, stained with nc82 (magenta) and DAPI (teal). **i** UAS-shits crossed to L40987-GAL4 trained by *D. ananassae* at the restrictive (left) and permissive (right) temperature. Blue lines indicate permissive temperature, while red lines indicate restrictive temperature. Trained and untrained states are labeled. Plots are standard Tukey Box Plots, where the bounds of the shaded box represent the first and third quartile; whiskers indicate the minimum and maximum data point within 1.5 times the interquartile range. Mean is denoted with an 'X'. Data used to generate the plots have been superimposed on the graph as open circles ($n = 12$ biologically independent experiments) (*$p < 0.05$, a → $p < 0.05$ comparing trained and untrained students within a given temperature when both restrictive and permissive temperatures are tested, b → $p = $ ns comparing trained and untrained students within a given temperature when both restrictive and permissive temperatures are tested). Insets in **b**, **d**, **f**, and **h** correspond to magnifications shown at right. All untrained states and comparisons are shown in Supplementary Fig. 16

identify three brain regions whose activation seems dispensable when using these particular genetic reagents. Following identification of these regions, we further elucidated a neuronal subset in three distinct regions—the antennal lobe, optic lobe, and the fan-shaped body—that are necessary for dialect training.

The current study has identified a novel role for an olfactory receptor, Or69a in conjunction with the ab9a OSN, where twin Ors are simultaneously expressed in the same OSN population that sense species-specific pheromones and food odorants[30] (Fig. 2). The OSN enervates the D glomerulus in the antennal lobe, which we hypothesize signals to the lateral horn and MB through interneurons. A cluster of lateral horn neurons, termed P1, has been identified to collect olfactory and contact chemosensory signals and subsequently elicits male courtship, and might be involved[57]. The question that arises for future work is how Or69a and projection neurons from its associated D glomerulus contribute excitatory input to circuitry of the lateral horn and the mushroom body.

Previous studies have highlighted the role of L1 and L2 motion-detecting neurons as specialized for the detection of moving light and dark edges, while L3 are involved in landmark detection and dark edge detection[39,40]. Additional work has identified the L4 neuron as involved in the connectome of motion recognition[45]. Using genetic agents restricted to L2 and L4 neurons, we observe that independent silencing of either neuron results in perturbed dialect acquisition (Fig. 3, Supplementary Fig. 18). Our results suggest that L4 might have a specific role in motion detecting of specific *Drosophila* movements. This finding is consistent with previous studies that suggest that L4 neurons may play a role in spatial summation and pooling information about contrast changes in motion detection[37,42,58,59]. L2 and L4 neurons make a diverse range of connections in the medulla. Knowing this, our data raise the possibility that downstream motion computations are distributed among many different neuron types. These neurons may then further converge in deeper layers of the visual system. This additional connectivity may tune neurons to particular features, termed the 'optic glomeruli'[60–64]. These more downstream and specialized neurons could then inform specific outputs appropriate to the visual stimulus during dialect training. It will be interesting to investigate the connectivity of L2 and L4 to these downstream neurons and extract what feature information is being detected (Supplementary Movies 1 and 2).

The current work also investigates the role of the well-defined fly brain region, the FB, and several groups of FB neurons. The results show that FB neurons participate in the consolidation of memory pertaining to social interactions. We identify a subgroup of large-field FB neurons contained in ventral layer 5 of the FB. Outputs of layer 5 neurons are necessary for dialect training (Supplementary Movies 3 and 4). Silencing this particular layer or the entire FB results in perturbed dialect training (Fig. 4). As one of the distinct parts of the central complex in the fly brain[47], the FB is important for multiple functions. Previous studies have identified visual feature recognition and sleep primarily relying on dorsal FB layers, while avoidance behaviors largely rely on ventral and middle layers of FB. We identify a ventral layer of the FB necessary for dialect training. The current state of FB research suggests that different FB layers serve as hubs for distinct behavioral information and our results suggest that layer 5 could be a mediator or consolidation point for social information.

Collectively, we present a model where inhibiting the activation of synaptic transmission in specific brain regions disrupts acquisition of salient information that takes place during a weeklong socialization period. This inhibition then, in turn, disrupts efficient dialect acquisition and/or fails to allow interspecies communication (Fig. 5, Supplementary Fig. 20). Given the need for multiple sensory inputs, dialect training is fundamentally different from previously described teacher–student paradigms that utilize exclusively visual cues as a means of information transfer. In addition, we suggest that this study also points to previously unappreciated functions of the *Drosophila* MB in integrating information from olfactory and visual inputs, perhaps through the L4 motion-detecting neuron in the optic lobe and the D glomerulus and ab9a OSN of the antennal lobe.

Dialect training between *D. melanogaster* and *D. ananassae* results in the ability of each species to more efficiently receive information about a common predator. It is important to note that the training period, during which socialization of two or more species takes place, is clearly a time when active learning is occurring. One hypothesis that may pertain to this particular phenomenon is the idea of perceptual learning, which is the process by which the ability of organisms to respond to environmental cues is improved through experience[65,66]. The experience, in this case, is the two species being cohoused, which leads to subsequent improved responses about environmental cues (fly communication).

It is not clear what specific information is being shared during socialization and cohabitation. Plasticity requires multiple circuits to function in unison to facilitate overcoming dialect barriers during teacher–student interactions. We do not know as of yet whether specific content is being shared during the training period, or whether socialization with another species simply makes for a permissive state where information is more efficiently perceived. This again points to the idea of perceptual learning,

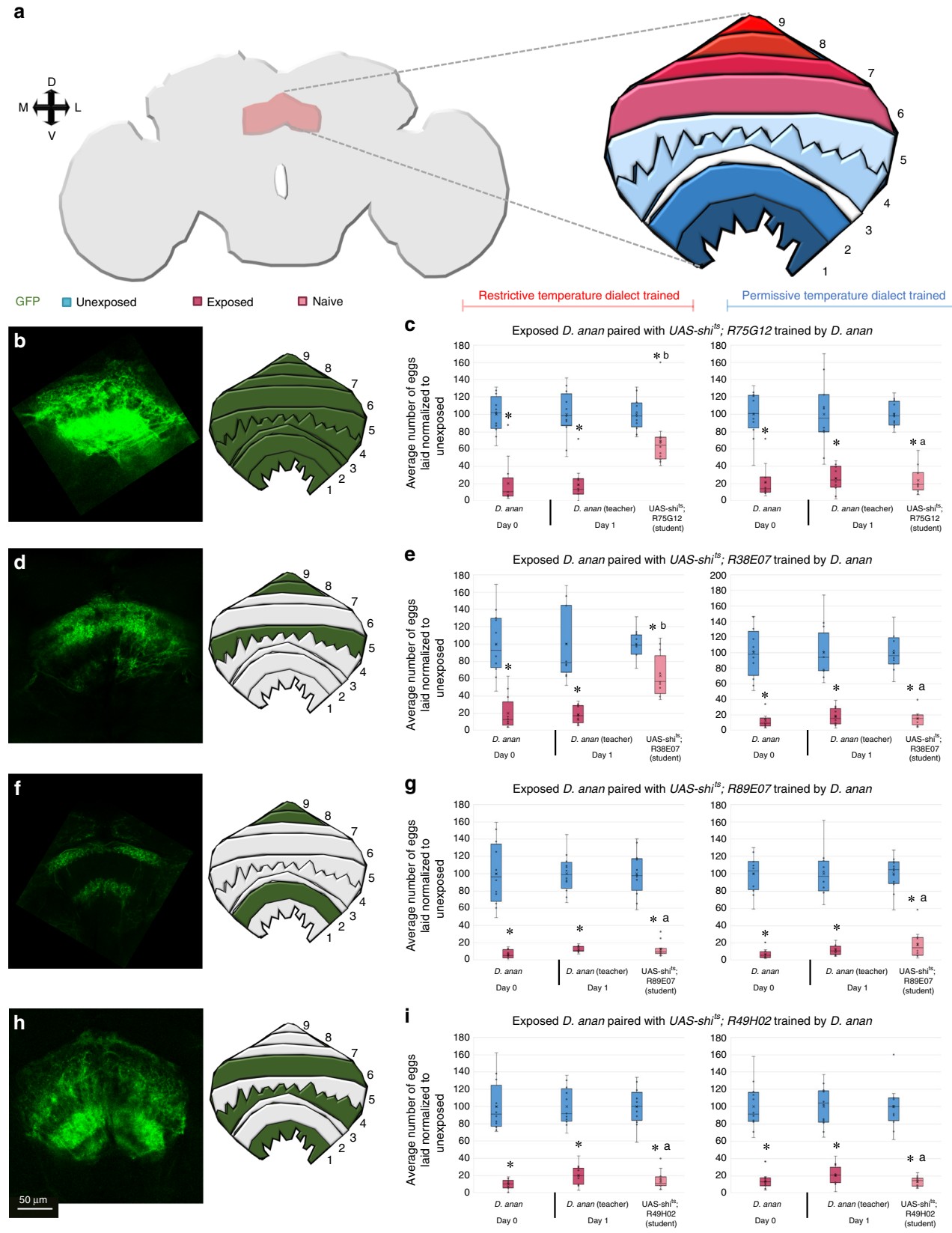

where this study is the first instance of such experience dependent enhancement of environmental responses in Drosophila. However, future studies are required to dissect and characterize interspecific differences in behaviors as a means to identify the experiences gained. An unsupervised analysis would be ideal,

given that so many unique sensory inputs are being utilized. This may then be followed with genetic and optogenetic dissection of the behaviors identified.

Our findings hint that some adult behaviors emerge only as a result of previous social experiences that are indeed species

**Fig. 4 Region 5 of the fan-shaped body is necessary for dialect training. a** Schematic of the *Drosophila* fan-shaped body (FSB) with the various regions (1–9) highlighted. **b** Expression pattern and schematic of the R75G12[GAL4] FSB driver shows pan-FSB marking. Percentage of eggs laid by exposed flies normalized to eggs laid by unexposed flies is shown at the permissive (22 °C) or restrictive (30 °C) temperature. **c** UAS-shi[ts] crossed to R75G12[GAL4] trained by *D. ananassae* at the permissive temperature shows wild-type trained state, but at the restrictive temperature shows defective acquisition in the trained state. **d** Expression pattern and schematic of the R38E07[GAL4] FSB driver shows FSB marking in regions 5, 8, and 9. **e** UAS-shi[ts] crossed to R38E07[GAL4] trained by *D. ananassae* at the permissive temperature shows wild-type trained state, but at the restrictive temperature shows defective acquisition in the trained state. **f** Expression pattern and schematic of the R89E07[GAL4] FSB driver shows FSB marking in regions 2, 8, and 9. **g** UAS-shi[ts] crossed to R89E07[GAL4] trained by *D. ananassae* at the permissive and the restrictive temperatures shows wild-type trained state. **h** Expression pattern and schematic of the R49H02[GAL4] FSB driver shows FSB marking in regions 2, 8, and 9. **i** UAS-shi[ts] crossed to R49H02[GAL4] trained by *D. ananassae* at the permissive and the restrictive temperatures shows wild-type trained state. All untrained states are shown in Supplementary Fig. 18. Blue lines indicate permissive temperature, while red lines indicate restrictive temperature. Trained and untrained states are labeled. Plots are standard Tukey Box Plots, where the bounds of the shaded box represent the first and third quartile; whiskers indicate the minimum and maximum data point within 1.5 times the interquartile range. Mean is denoted with an 'X'. Data used to generate the plots have been superimposed on the graph as open circles ($n = 12$ biologically independent experiments) (*$p < 0.05$, a → $p < 0.05$ comparing trained and untrained students within a given temperature when both restrictive and permissive temperatures are tested, b → $p$ = ns comparing trained and untrained students within a given temperature when both restrictive and permissive temperatures are tested. Comparison groups are found in Supplementary Fig. 18)

specific. For example, cohabitation of *D. melanogaster* with *D. ananassae* does increase the efficiency of these two species to communicate, but it does not enhance the *D. melanogaster*'s ability to communicate with *D. willistoni*[15]. When cohoused with two difference species, *D. melanogaster* can then more efficiently learn from each of these two different species, indicating that barriers from two different dialects can be simultaneously overcome in the same animal[15]. Thus, this supports the idea that specific information may be shared during the cohabitation and socialization period. Such multi-species experiences take place in nature, where relevant ecological pressures are ever present and multiple species co-exist and compete for resources while simultaneously cooperate against common foes[67]. Thus, unlike laboratory conditions that typically seek to grow animals either in social isolation and/or monocultures, wild individuals in the wild must constantly weigh the costs of competition against the benefits of cooperation with every intra- and inter-species interaction. Cognitive plasticity wields a real fitness benefit to the individual and the group, where sharing of information directly, or by coincident bystanders, could result in behavioral immunity to pan and specific threats. Our observations also raise the interesting possibility that different species share information not just about common predators, but other environmental inputs as well, such as the social environment. Perhaps even rudimentary social learning neuronal circuitry can be co-opted to receive and integrate useful information from a variety of close and distantly related species.

We have only just arrived at the boundary between insect behavioral neuro-genetics, evolution and ecology, and sociobiology. Our study integrates the vast tool-box and the life history of the Drosophila model in an effort to dissect the neural circuitry governing social behavior.

## Methods

**Insect species/strains**. The *D. melanogaster* strain Canton-S (CS) was used as the wild-type strain and used for outcrosses. The Drosophila species *D. ananassae* and *D. virilis* were acquired from the Drosophila Species Stock Center (DSSC) at the University of California, San Diego, stock numbers 14024-0371.13 and 15010-1051.87, respectively. *L1*, *L2*, *L3*, *L4^0987*, and splitL4[GAL4] lines were kindly provided by Marion Silies (European Science Institute, Germany). The CD8-GFP line was kindly provided by Mani Ramaswami (Trinity College, Dublin, Ireland). All stocks used in experiments are listed in Table S1 with stock numbers shown (when applicable).

Flies aged 3–6 days post-eclosion on fresh Drosophila media were used in all experiments. All flies were maintained at 22 °C (the permissive temperature of experiments described later) with ~30–45% humidity, with a 12:12 light:dark cycle at light intensity 16[7] with 30–55% humidity dependent on weather. Light intensity was measured using a Sekonic L-308DC light meter. The light meter utilized measures incident light and is set at shutter speed 120, sensitivity at iso8000, with a 1/10 step measurement value (f-stop). These are conditions used in previous

studies[12–15]. All species and strains used were maintained in fly bottles (Genesse catalog number 32-130) containing 50 mL of standard Drosophila media. Bottles were supplemented with three Kimwipes rolled together and placed into the center of the food to promote pupation. *Drosophila* media was also scored to promote oviposition. Fly species stocks were kept separate to account for visual cues that could be conferred if the stocks were kept side-by-side.

We utilized the generalist Figitid larval endoparasitoid *Leptopilina heterotoma* (strain Lh14), that is known to infect a wide array of Drosophilids[68–70]. *L. heterotoma* strain Lh14 originated from a single female collected in Winters, California in 2002. To propagate wasp stocks, we used adult *D. virilis*, which were placed in batches of 40 females and 15 males per vial (Genesse catalog number 32-116). The strain we used has been maintained on *D. virilis* since 2013, though was originally maintained on *D. melanogaster*. Adult flies are allowed to lay eggs in these standard Drosophila vials that contain 5 mL standard Drosophila media supplemented with live yeast (~25 granules) for 4–6 days. Flies were then replaced by adult wasps—15 female and 6 male wasps—for infections. Infection timing gives the wasps access to the L2 stage of *D. virilis* larvae. Vials that contain wasps are supplemented with ~500 μL of a 50% honey/water solution that is applied to the inside of the cotton vial plugs. The honey used was organic, raw and unfiltered. Wasps aged 3–7 days post-eclosion were used for all infections and experiments. Wasps were never reused for experiments, nor were they used for stock propagation if used for experiments.

**Fly duplexes**. We utilized the previously described fly duplex to examine teaching ability following dialect training[13,15]. The fly duplexes were constructed (Desco, Norfolk, MA) by using three 25 mm × 75 mm pieces of acrylic that were adhered between two 75 mm × 50 mm × 3 mm pieces of acrylic via clear acrylic sealant. This yields two compartments separated by one 3-mm-thick acrylic piece. Following sealant curing, each duplex is soaked in water and Sparkleen detergent (Fisherbrand™ catalog number 04-320-4) overnight, then soaked in distilled water overnight and finally air-dried.

For experiments using fly duplexes (teacher–student interaction), bead boxes (6 slot jewelers bead storage box watch part organizer sold by FindingKing) were used to accommodate 12 replicates of each treatment group. Compartments measure 32 × 114 mm with the tray in total measuring 21 × 12 × 3.5 mm. Each compartment holds 2 duplexes, and the tray holds a total of 12 duplexes (accounting for the 12 replicates). When setting up experiments, empty duplexes are placed into the bead box compartments. Fifty milliliters of standard *Drosophila* media in a standard Drosophila bottle (Genesse catalog number 32-130) is heated (Panasonic brand microwave) for 54 s. This heated media is then allowed to cool for 2 min on ice before being dispensed into the duplexes. Each duplex unit is filled with ~5 mL of the media and further allowed to cool until solidification. Food was scored to promote oviposition. We find that *D. ananassae* oviposition rate is extremely low in control conditions at 22 °C without scoring. The open end of the Fly Duplex is plugged with a cotton plug (Genesse catalog number 51-102B) to prevent insect escape. Ten female flies and 2 male flies are placed into one chamber of the Fly Duplex in the control, while 20 female Lh14 wasps are placed into the neighboring chamber in the experimental setting for 24 h. After the 24-h exposure, flies and wasps were removed by anesthetizing flies and wasps in the fly duplexes. Control flies undergo the same anesthetization protocol. Wasps are removed and replaced with ten female and two male 'student' flies. All flies are placed into new, clean duplexes for the second 24-h period, containing 5 mL *Drosophila* media in a new bead box, prepared in the same manner as described for day 0 (wasp exposure period). Plugs used to keep insects in the duplexes are replaced every 24 h to prevent odorant deposition on plugs that could influence behavior. The oviposition bead box from each treatment is also replaced 24 h after the start of the experiment, and the second bead box is removed 48 h after the start of the experiment. Fly egg

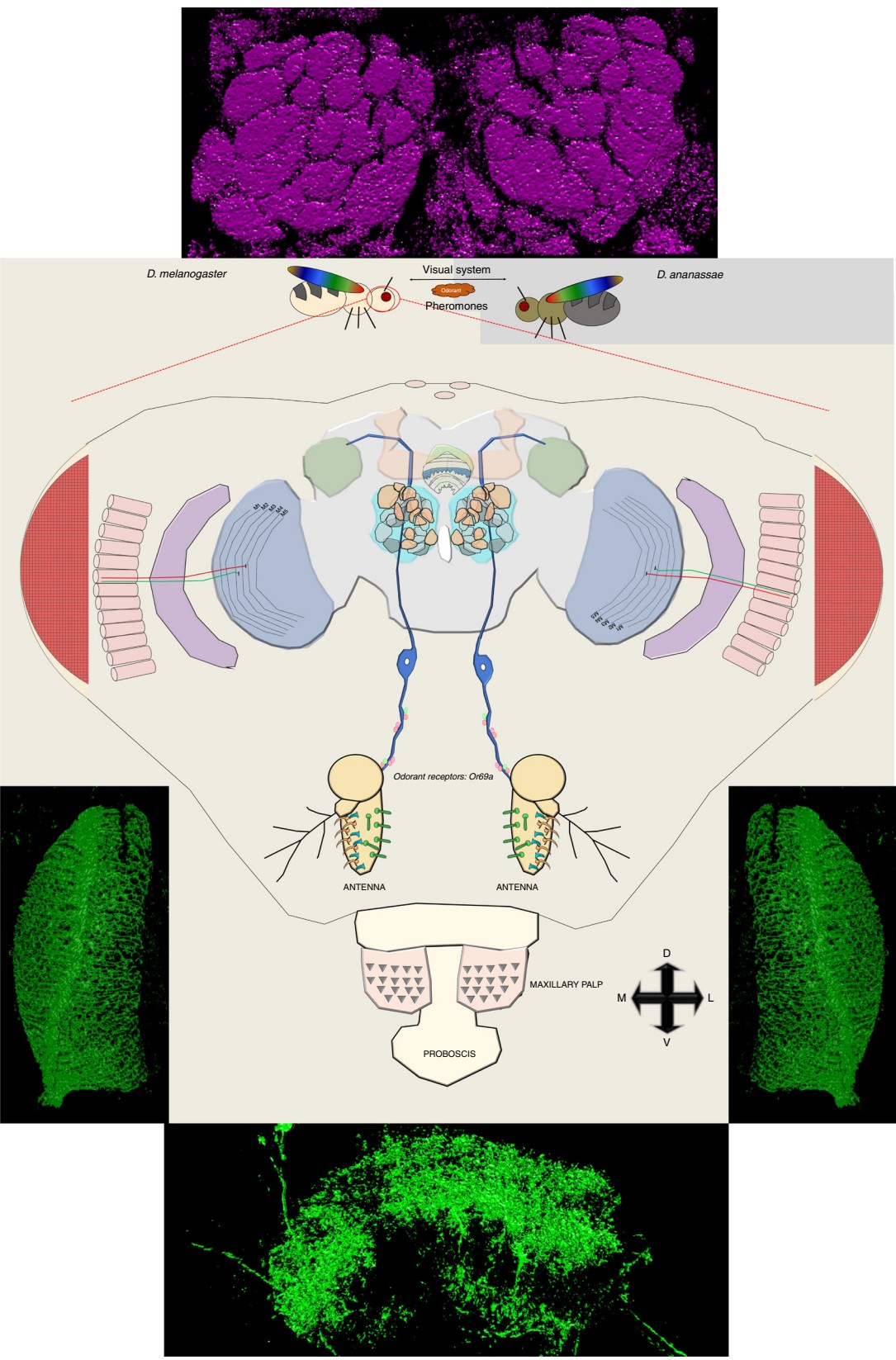

counts from each bead box were made at the 0–24 and 24–48-h time points in a blinded manner. Experimental conditions were coded and eggs were scored. The coder and counter were always two different individuals to maintain blinded counting. This way, the counter was unaware of both genotype and treatment. Decoding of the data was performed once an experiment was fully completed by the original coder. This maintained a blinded study until the conclusion of testing. In addition, the initial neuron screen (Supplementary Figs 2–14) fly lines were coded prior to generating crosses, such that the counter was unaware of what possible genotypes may be being utilized. Used bead boxes and fly duplexes are soaked overnight in a 4% Sparkleen detergent solution (Fisherbrand™ catalog number 04-320-4), rinsed with distilled water, and allowed to air dry before subsequent use.

All experimental treatments as described above were run at 22 °C with a 12:12 light:dark cycle at light intensity of $16_7$, measured using a Sekonic L-308DC light

**Fig. 5** Model for the neural circuitry of dialect training in Drosophila. Inter-species communication and dialect learning is dependent on the presence of multiple cues between *D. melanogaster* and *D. ananassae*. These cues are, in part, olfactory, visual, and ionotropic in nature. In *D. melanogaster*, we have identified multiple brain regions and neurons that contribute to the dialect training. Regions we identified to be involved in dialect training—the optic lobe (blue), mushroom body (orange), antennal lobe (teal), lateral horn (green), fan-shaped body (green/white), and ellipsoid body (light green). For the antennal lobe, we identify the D glomerulus, innervated by *Or69a* olfactory receptor neuron (ORN). For the fan-shaped body, we identify region 5 as necessary for dialect training. For the optic lobe we identify the motion-detecting circuit as necessary by isolation of the L4 neurons. Flanking the model, super-resolution microscopy with a 3D render is provided of the key regions of the fly brain that were investigated—region 5 of the fan-shaped body, the L4 neurons, and the antennal lobe. Super-resolution images without 3D render are located in Supplementary Fig. 20

meter, using 12 replicates at 40% humidity unless otherwise noted. Fly duplexes and bead boxes soaked a 10% Sparkleen solution after every use for 4 h at minimum and subsequently rinsed with distilled water and air-dried overnight. All egg plates were coded and scoring was blind as the individual counting eggs was not aware of treatments or genotypes. Genotypes were numerically coded and not revealed until the conclusion of the experiment.

**Dialect training**. Drosophila species were cohoused during dialect training in standard Drosophila bottles (Genesee catalog number 32-130) containing 50 mL standard Drosophila media. Three Kimwipes were rolled together and placed into the center of the food. Three bottles were prepared per treatment. *D. melanogaster* and *D. ananassae* are incubated in each bottle with 100 female and 20 males of each species per bottle. Every 2 days, flies are placed into new bottles that prepared in the identical manner described above. Flies were cohabitation for ~168 h (7 days). This cohabitation takes place at either 30 or 22 °C depending on treatment parameters. Following cohabitation, flies are anesthetized and the two species are separated. *D. melanogaster* are then used as students to wasp or mock exposed *D. ananassae* teachers. This portion of the experiment always took place at 22 °C. For example, we cohoused *D. melanogaster* and *D. ananassae* for 1 week at either 22 or 30 °C. Following the weeklong cohabitation, we separated the dialect-trained flies. Dialect-trained *D. melanogaster* were placed in duplexes next to *D. ananassae* either mock or wasp exposed exclusively at 22 °C. These *D. ananassae* had not encounted *D. melanogaster* until this point of the experiment.

Oviposition rate was used to determine the degree of communication, being one of three possible categories: 'no communication' (no statistical difference in oviposition rate), 'partial communication' (decreased oviposition but not less than 50% of unexposed), or 'full communication' (statistically decreased oviposition beyond 50% of unexposed).

All experimental, dialect training treatments were run either at 30 or 22 °C (as indicated in figures) with a 12:12 light:dark cycle at light intensity $16_7$, using 12 replicates at 40% humidity unless otherwise noted. Light intensity was measured using a Sekonic L-308DC light meter. The light meter measures incident light and was set at shutter speed 120, sensitivity at iso8000, with a 1/10 step measurement value (f-stop). In addition, all treatments were coded and scoring was blind as the individual counting eggs was not aware of treatments or genotypes. All raw egg counts and corresponding *p*-values are provided in supplementary Data 1.

**Cross setup**. For experiments utilizing UAS-shi^ts, male flies containing tissue specific drivers were crossed to virgin female flies containing the UAS construct of interest. Twenty female UAS virgin flies were mated to 10 males containing the selected hairpin/construct. These constructs were coded such that the experimenter was not aware of the lines being tested. Crosses were performed in standard Drosophila fly bottles (Genesse catalog number 32-130) containing 50 mL of standard Drosophila media, supplemented with 10 granules of activated yeast. Crosses were kept at ~20 °C, with a 12:12 light:dark cycle at light intensity $16_7$ with 30–45% humidity dependent on weather. Crosses were moved to new bottles every 2 days until oviposition rates declined, at which point the adults were disposed of. F1's were harvested upon eclosion and stored in standard Drosophila vials (Genesse catalog number 32-116) containing 5 mL Drosophila media with approximately 20 female and 5 male flies until utilized for experimentation.

**Immunofluorescence**. Drosophila brains containing genetically distinct GAL4 constructs driving a mCD8-GFP were dissected in a manner previously described[12,13]. Flies expressing GAL4 in a tissue specific manner driving mCD8-GFP were placed in batches into standard vials (Genesee catalog number 32-116) of 20 females, 2 males. Three vials were prepared to produce three replicates to account for batch effects of expression. We observed no expression batch effects. Brains that were prepared for immunofluorescence were fixed in 4% methanol-free formaldehyde in PBS with 0.001% Triton-X for ~5 min. The samples were then washed in PBS with 0.1% Triton-X and placed into block solution for 2 h (0.001% Triton-X solution 5% normal goat serum). Samples were then placed into a 1:10 dilution of nc82 antibody (Developmental Studies at Hybridoma Bank, University of Iowa, Registry ID AB 2314866) to 0.001% Triton-X overnight at 4 °C. Following the overnight primary stain, samples were washed in PBS with 0.1% Triton-X three times. Secondary Cy3 antibody (Jackson Immunoresearch) was then placed on samples in a 1:200 ratio in 0.001% Triton-X. Following secondary staining, samples

were washed in PBS with 0.1% Triton-X three times. This was followed by a 10-min nuclear stain with 4′, 6-diamidino-2-phenylindole (DAPI) with three additional washes in PBS with 0.1% Triton-X before being mounted in Vecta shield (vector laboratories catalog number: H-1000). Samples were then immediately imaged.

**Imaging**. A Zeiss LSM 880 with Airyscan Confocal microscope was used for brain imaging for all samples with the exception of Supplementary Fig. 8, where a Leica SP8 confocal microscope was used with Leica lighting image deconvolution. Image averaging of 16x during image capture was used for all images. Whole head images of antennal GFP expression were taken using Nikon E800 Epifluorescence microscope with Olympus DP software. For Fig. 5, Supplementary Movies 1–4, and Supplementary Fig. 20, a spectral confocal/super-resolution microscope LSM 880 with Airyscan was used.

**Statistics and reproducibility**. Statistical tests on exposed vs unexposed/teacher vs student interactions were performed. In addition, we performed comparisons between trained and untrained states of students within a given temperature. We are able to perform this comparison as these experiments were performed in parallel. For example, restrictive temperature trained and untrained students were compared to see if training had an effect. In figures, we denote these changes with letters, where 'a' means that a statistically significant difference was found between the trained and untrained groups in addition to the trained group being different from the 50% threshold; 'b' meaning that there was no difference in trained and untrained states in the restrictive temperature; and 'c' indicates no difference in trained and untrained states in the permissive temperature. All statistical comparisons were performed in Microsoft Excel. Welch's two-tailed *t*-tests were performed for all data comparisons. *P*-values reported were calculated for comparisons between paired treatment group and unexposed and are included in supplemental data 1. All behavioral experiments were comprised of 12 replicates. Each replicate was an independent wasp exposure, and utilize different trained individuals. Plots are standard Tukey Box Plots, where the bounds of the shaded box represent the first and third quartile; whiskers indicate the minimum and maximum data point within 1.5 times the interquartile range. Mean is denoted with an 'X'. Data used to generate the plots have been superimposed on the graph as open circles.

**Reporting summary**. Further information on research design is available in the Nature Research Reporting Summary linked to this article.

## Data availability

All raw egg counts to corresponding figures are reported in Supplementary Data 1. No restrictions apply to access of all the data. Flies used are reported in Supplementary Table 1 and are available upon request.

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

## Acknowledgements

We thank Mani Ramaswami, Marion Silies, FlyBase, the Drosophila Species Stock Center, and the Bloomington Drosophila Stock Center, for stocks. We acknowledge grants from Geisel School of Medicine at Dartmouth, the National Institute of Health Pioneer grant 1DP1MH110234 (GB), and the Defense Advanced Research Projects Agency grant HR0011-15-1-0002 (G.B.).

## Author contributions

B.Z.K., S.H., and J.B. maintained fly and wasp stocks; B.Z.K., S.H., and J.B. performed immunofluorescence and imaging; B.Z.K., S.H., J.B., and G.B. designed and performed experiments, data curation, and statistical analyses; B.Z.K. and G.B. wrote the paper; B.Z. K., J.B., and G.B. revised the paper; G.B. supervised this project.

## Additional information

**Competing interests:** The authors declare no competing interests.

