## [Peer Review File · Communications Biology]

Reviewers' comments:

Reviewer #1 (Remarks to the Author):

This report describes insight into the operational neural circuitry that governs a social behavior apparent across different species of Drosophilids. The manuscript is well written, the experiments are straightforward and the results are compelling, and the subject matter will capture the interests of many different biologists. The work is significant as it challenges the "asocial" notion of Drosophila and employing the genetic tools in this organism, this manuscript offers mechanistic insight into the sensory systems that underlie a biologically relevant social learning task.

The majority of my criticisms are minor and most center on alternative methods of presentation to aide the reader. I do advise that the analyses be reconsidered prior to publication – so that direct comparisons between the groups of interest can be made (permissive vs. restrictive, naïve vs. trained), since as presented – those comparisons cannot be supported by the statistical tests chosen, and are reliant on statistical inferences rather than direct comparisons.

Comments - While the paper is clear and well written – I think the introduction is needlessly long and could be condensed. The discussion on honeybees is interesting but perhaps not germane to understand the article.

The figures are not presented in an intuitive fashion. I suggest the authors switch the order of the permissive and restrictive temperatures and consider color coding those panels (blue and red or something to that effect).

The word naïve – as the description for the "experiment" is confusing – especially since it makes one think the animals have not been tested which the term is used elsewhere. Maybe label those results as EXPERIMENTAL – to get across the idea that these truly what the assay is testing for...

I also wonder if the statistics are the best – in many regards I think the authors are under-reporting the effect. The comparison of interest is not necessarily the ones between the unexposed and exposed for the trained D. mel – but rather the difference between the untrained and trained flies – and between the permissive and restrictive temperatures. I can appreciate the fact that the authors are aware that they cannot make direct comparisons between these as they have their "own" controls – but they do make a point that this is the comparison of interest throughout the report. So, if I understand the experimental design completely, the unexposed controls are different sets of animals – and "pairing" of those controls to an experiment is not truly paired but rather represents an independent set of measures. If that is the case, than it would be perfectly reasonable to reorganize the data and perform an ANOVA and make the direct comparisons between trained and untrained and between the permissive and restrictive temperatures. As it stands now, the authors make statistical inferences between the values of these measure but cannot truly make definitive statements because there are no statistical tests of those comparisons.

Figure 1 D and E show a map of different brain regions and are illustrative of data presented in the supplemental files. Please remove these panels – or include an abbreviated set of figures from the panels – see comment above about statistical analysis – one could make direct comparisons between different GAL4 lines using an ANOVA test.

Another concern is there appear to be little to no control for different genetic backgrounds of the mutant and RNAi lines. For all the UAS-shi experiments – this isn't an issue as the permissive temp is the ideal genetic control. The data presented that implicate the OR69a though could be a product of different genetic backgrounds and it would be nice to see some genetic controls for these experiments in the supplemental – for example the parental lines (OR69a –GAL4, or the UAS-RNAi – or ideally – the progeny of the OR69a-GAL4 crossed to a UAS-GFP RNAi line.

The authors should also temper the language regarding "dispensable for dialect training" throughout the manuscript and figures. Negative results, including the ones in this report are difficult to interpret.

Considering the authors didn't try (which is perfectly reasonable and am not suggesting they try) and establish that the genetic reagents (UAS-shi) is equally effective in multiple cell lineages – then I would argue that there could be technical rather than biological reasons. I would be a little more circumspect and state something to the effect of "We found no evidence utilizing our reagents, that the following brain regions are involved in dialect learning...."

Minor comments :

There are words that are hyphenated that shouldn't be e.g., Non-caste-based, tumor-genesis, mate-choice. This probably is a formatting issue.

The lines in the text (bottom of page five) and figure legend ... "Interspecies dialect learning, unlike intraspecies social learning, is multimodal, requiring, at minimum, olfactory, visual, sex-specific, temporal, neuronal, and ionotropic cues for successful dialect acquisition" need to be altered. What is an ionotropic cue? Or neuronal cue? Omit these as these aren't really "sensory" systems.

Not sure what value of the supplemental movies add to the manuscript.

Reviewer #2 (Remarks to the Author):

In this very interesting and solid paper, Kacsoh and colleagues investigate the neurobiological basis of the circuit that mediates social learning between *Drosophila* species, focusing on visual and olfactory circuits. Using behavioural and genetic tools, the authors identify *Drosophila* brain regions important for communication, including particular neuron located in these regions.

I have a theoretical point that I think should be modified (referring to group selection, see below).

Overall, I would like to congratulate with the authors for this work, and to suggest a few minor changes to improve the clarity of the manuscript.

- I do not agree with the group selection interpretation of the results : "Our study integrates the vast tool-box and the life history of the *Drosophila* model in an effort to dissect the neural circuitry governing social behavior that suggests fitness of the individual is not the primary determinant of fitness in a natural setting. The sociobiology trait of a group is not determined by a group member's genetic fitness, but by the summed effects of all of the survival indices of the group." Ultimately, if one individual A is not able to take advantage of communication from other individuals, A will have a net individual disadvantage. In any case, I do not think this research shows much about the group selection and I would omit this interpretation not supported by the data.

other comments:

- I wonder whether these effects of improved transmission of information after exposure can be conceptualised as perceptual learning. This might expand the impact of this work.

- In subsequent studies, it would be interesting to use unsupervised behavioural analysis to characterise the interspecific differences in behaviour that might trigger the teaching (e.g. Klibaite, U., Berman, G. J., Cande, J., Stern, D. L. & Shavitz, J. W. An unsupervised method for quantifying the behavior of paired animals. *Phys. Biol.* 14, (2017).)

Minor:

- Instead of "asocial" fruit fly it is probably more technical to speak about "solitary"

- Typo ananassae...

- In the introduction the word "demonstrate" "demonstration" etc is used quite often. In life sciences this can be often be substituted with "show" "suggest" "indicate" etc.

- Previous data has \diamond have

- Italics for the name of the species, where not used

- P- 25 "coding/decoding" : probably better to explain in what this procedure consists on the first occurrence

- What is the light intensity?
- Dialect training and other cases: I think species are "co-housed" not "co-habitated" (places are habituated, not species)
- Dialect training: inconsistent use of tenses (present, past)
- Statistical analysis: "v" 0 "vs"?
- References: several typos, some refernces do not appear in the text (e.g. CHittka, Niven)
- Figures:
 - - Danan... ♦ D. anan and similar cases (e.g. 1D)
 - - - in many case the font is way too small
- PLease read again all the captions of the figures and supplementary materials because I am travelling and I cannot read them on paper at this time

Reviewers' comments:

Reviewer #1 (Remarks to the Author):

This report describes insight into the operational neural circuitry that governs a social behavior apparent across different species of Drosophilids. The manuscript is well written, the experiments are straightforward and the results are compelling, and the subject matter will capture the interests of many different biologists. The work is significant as it challenges the "asocial" notion of Drosophila and employing the genetic tools in this organism, this manuscript offers mechanistic insight into the sensory systems that underlie a biologically relevant social learning task.

The majority of my criticisms are minor and most center on alternative methods of presentation to aide the reader. I do advise that the analyses be reconsidered prior to publication – so that direct comparisons between the groups of interest can be made (permissive vs. restrictive, naïve vs. trained), since as presented – those comparisons cannot be supported by the statistical tests chosen, and are reliant on statistical inferences rather than direct comparisons.

Comments - While the paper is clear and well written – I think the introduction is needlessly long and could be condensed. The discussion on honeybees is interesting but perhaps not germane to understand the article.
We have condensed the introduction by removing extraneous sections.

The figures are not presented in an intuitive fashion.

We have reorganized the order of the figures, in addition to changing panels and adding greater explanation for temperature (permissive/restrictive) points.

I suggest the authors switch the order of the permissive and restrictive temperatures and consider color coding those panels (blue and red or something to that effect).

Added colored trim around each permissive (blue) and restrictive (red) temperature experiments. For experiments that are not conditional, but were run at 22C, we maintain the blue color coding for clarity. Additionally, we have provided more explanation in the methods.

The word naïve – as the description for the “experiment” is confusing – especially since it makes one think the animals have not been tested which the term is used elsewhere. Maybe label those results as EXPERIMENTAL – to get across the idea that these truly what the assay is testing for...

Thank you for highlighting this point. We agree with the reviewer that this is a confusing series of terms. We have changed ‘naïve’ to ‘untrained’ while trained has been altered to ‘dialect trained.’ We hope that this change provides more clarity to the terminology. In instances where "naïve" is used we restrict this to mean flies that have not encountered or otherwise exposed to wasp.

I also wonder if the statistics are the best – in many regards I think the authors are under-reporting the effect. The comparison of interest is not necessarily the ones between the unexposed and exposed for the trained D. mel – but rather the difference between the untrained and trained flies – and between the permissive and restrictive temperatures. I can appreciate the fact that the authors are aware that they cannot make direct comparisons between these as they have their “own” controls – but they do make a point that this is the comparison of interest throughout the report. So, if I understand the experimental design completely, the unexposed controls are different sets of animals – and “pairing” of those controls to an experiment is not truly paired but rather represents an independent set of measures. If that is the case, than it would be perfectly reasonable to reorganize the data and perform an ANOVA and make the direct comparisons between trained and untrained and between the permissive and restrictive temperatures. As it stands now, the authors make statistical inferences between the values of these measure but cannot truly make definitive statements because there are no statistical tests of those comparisons.

We performed additional comparisons between trained and untrained states of flies within a given temperature. We are able to perform this comparison as these experiments were performed in parallel. For example, restrictive temperature trained and untrained students were compared to see if training had an effect. In figures, we denote these changes with letters, where ‘a’ means that a statistically significant difference was found between the trained and untrained groups in addition to the trained group being different from the 50% threshold; ‘b’ meaning that there was no difference in trained and untrained states in the restrictive temperature; and ‘c’ indicates no difference in trained and untrained states in the permissive temperature. We have indicated these changes in the figures, legends, and methods section of the

manuscript. In each case the statistical test used and the specific groups being compared is detailed in the figure legends.

Figure 1 D and E show a map of different brain regions and are illustrative of data presented in the supplemental files. Please remove these panels – or include an abbreviated set of figures from the panels – see comment above about statistical analysis – one could make direct comparisons between different GAL4 lines using an ANOVA test.

We have removed these panels and placed them into the supplemental summary section.

Another concern is there appear to be little to no control for different genetic backgrounds of the mutant and RNAi lines. For all the UAS-shi experiments – this isn't an issue as the permissive temp is the ideal genetic control. The data presented that implicate the OR69a though could be a product of different genetic backgrounds and it would be nice to see some genetic controls for these experiments in the supplemental – for example the parental lines (OR69a-GAL4, or the UAS-RNAi – or ideally – the progeny of the OR69a-GAL4 crossed to a UAS-GFP RNAi line.

We have performed additional tests of genetic background lines. We highlight that figure 1 C-D has UAS-shi^{ts} outcrossed to wild-type lines and is tested at both temperatures to be used. Additionally, we provide Or69a-Gal4, UAS-Or69a^{RNAi}, and Or69a⁷⁻ data for outcrossed lines (construct/+) shown in Supplementary figure 16. These lines behave as wild-type, thus, we believe that our results are not due to genetic background effects.

The authors should also temper the language regarding “dispensable for dialect training” throughout the manuscript and figures. Negative results, including the ones in this report are difficult to interpret. Considering the authors didn't try (which is perfectly reasonable and am not suggesting they try) and establish that the genetic reagents (UAS-shi) is equally effective in multiple cell lineages – then I would argue that there could be technical rather than biological reasons. I would be a little more circumspect and state something to the effect of “We found no evidence utilizing our reagents, that the following brain regions are involved in dialect learning...”

Thank you for this suggestion. We do appreciate the limitations of negative results. We have changed our language to reflect the limitations of the reagents used, and we added a caveat that the efficiency of our genetic tools (e.g. UAS -shi) likely varies from cell to cell or brain region to another and therefore may produce different results.

Minor comments :

There are words that are hyphenated that shouldn't be e.g., Non-caste-based, tumor-genesis, mate-choice. This probably is a formatting issue.

We have corrected hyphenation issues in addition to some typos.

The lines in the text (bottom of page five) and figure legend ... “Interspecies dialect learning, unlike intraspecies social learning, is multimodal, requiring, at minimum, olfactory, visual, sex-specific, temporal, neuronal, and ionotropic cues for successful dialect acquisition” need to be altered. What is an ionotropic cue? Or neuronal cue? Omit these as these aren't really “sensory” systems.

Thank you for this clarification. We have removed these inputs.

Not sure what value of the supplemental movies add to the manuscript.

We suggest that the super-resolution videos of the brain regions identified may provide a starting point for future connectomics studies, but can remove them if the reviewers feel they are unnecessary.

Reviewer #2 (Remarks to the Author):

In this very interesting and solid paper, Kacsoh and colleagues investigate the neurobiological basis of the circuit that mediates social learning between *Drosophila* species, focusing on visual and olfactory circuits. Using behavioural and genetic tools, the authors identify *Drosophila* brain regions important for communication, including particular neuron located in these regions.

I have a theoretical point that I think should be modified (referring to group selection, see below). Overall, I would like to congratulate with the authors for this work, and to suggest a few minor changes to improve the clarity of the

manuscript.

- I do not agree with the group selection interpretation of the results : “Our study integrates the vast tool-box and the life history of the *Drosophila* model in an effort to dissect the neural circuitry governing social behavior that suggests fitness of the individual is not the primary determinant of fitness in a natural setting. The sociobiology trait of a group is not determined by a group member’s genetic fitness, but by the summed effects of all of the survival indices of the group.” Ultimately, if one individual A is not able to take advantage of communication from other individuals, A will have a net individual disadvantage. In any case, I do not think this research shows much about the group selection and I would omit this interpretation not supported by the data.

We have removed this section from the manuscript.

other comments:

- I wonder whether these effects of improved transmission of information after exposure can be conceptualised as perceptual learning. This might expand the impact of this work.

We have included a section on perceptual learning:

“Dialect training between *D. melanogaster* and *D. ananassae* results in the ability of each species to more efficiently receive information about a common predator. It is important to note that the training period, during which socialization of two or more species takes place, is clearly a time when active learning is occurring. One hypothesis that may pertain to this particular phenomenon is the idea of perceptual learning, which is the process by which the ability of organisms to respond to environmental cues is improved through experience (Gold and Watanabe 2010, R46-8; Sasaki et al. 2010, 53) . The experience, in this case, is the two species being co-housed, which leads to subsequent improved responses about environmental cues (fly communication).”

- In subsequent studies, it would be interesting to use unsupervised behavioural analysis to characterise the interspecific differences in behaviour that might trigger the teaching (e.g. Klibaite, U., Berman, G. J., Cande, J., Stern, D. L. & Shaevitz, J. W. An unsupervised method for quantifying the behavior of paired animals. *Phys. Biol.* 14, (2017).)

Thank you for this suggestion. We agree that this would be a very exciting avenue of future work and have added in a section in the discussion to reflect this.

“It is not clear what specific information is being shared during socialization and cohabitation. Plasticity requires multiple circuits to function in unison to facilitate overcoming dialect barriers during teacher-student interactions. We do not know as of yet whether specific content is being shared during the training period, or whether socialization with another species simply makes for a permissive state where information is more efficiently perceived. This again points to the idea of perceptual learning, where this study is the first instance of such experience dependent enhancement of environmental responses in *Drosophila*. However, future studies are required to dissect and characterize interspecific differences in behaviors as a means to identify the experiences gained. An unsupervised analysis would be ideal, given that so many unique sensory inputs are being utilized (Klibaite et al. 2017, 015006) . This may then be followed with genetic and optogenetic dissection of the behaviors identified (Cande et al. 2018, e34275) .”

Minor:

- Instead of “asocial” fruit fly it is probably more technical to speak about “solitary”

We have corrected this in the introduction.

- Typo *ananassae*...

Thank you for finding this typo. We have corrected this point.

- In the introduction the word “demonstrate” “demonstration” etc is used quite often. In life sciences this can be often be substituted with “show” “suggest” “indicate” etc.

We have changed the overuse of ‘demonstrate’ to other synonyms. Thank you for pointing this out. We feel that the changes allow the paper to be more easily read.

- Previous data has \diamond have

Thank you for finding this. We have corrected it.

- Italics for the name of the species, where not used
We have italicized all instances of species, when listed.
- P- 25 “coding/decoding” : probably better to explain in what this procedure consists on the first occurrence
We have elaborated on the coding/decoding processes in the methods.
- What is the light intensity?
We have indicated light intensity in the methods—how we measured, with what, and what the numbers mean.
- Dialect training and other cases: I think species are “co-housed” not “co-habitated” (places are habituated, not species)
We have changed all instances to co-housed.
- Dialect training: inconsistent use of tenses (present, past)
We have corrected tense changes
- Statistical analysis: “v” 0 “vs”?
We have corrected v to vs.
- References: several typos, some references do not appear in the text (e.g. CHittka, Niven)
We have corrected the references.
- Figures:
 - - Danan... ◊ D. anan and similar cases (e.g. 1D)
We have corrected these instances in the figures.
 - - - in many case the font is way too small
We have changed font size to suggested size based on journal specifications.
- Please read again all the captions of the figures and supplementary materials because I am travelling and I cannot read them on paper at this time
We have rechecked the legends.

REVIEWERS' COMMENTS:

Reviewer #1 (Remarks to the Author):

The authors adequately addressed all of my concerns in their revised manuscript and I would recommend publication of this elegant work.

Reviewer #2 (Remarks to the Author):

Thanks, all my doubts and requests of clarifications have been fully addressed.